# Anticancer Effects of the *Corchorus olitorius* Aqueous Extract and Its Bioactive Compounds on Human Cancer Cell Lines

**DOI:** 10.3390/molecules26196033

**Published:** 2021-10-05

**Authors:** John Paul Sese Tosoc, Olga Macas Nuñeza, Thangirala Sudha, Noureldien H. E. Darwish, Shaker A. Mousa

**Affiliations:** 1Department of Biological Sciences, College of Science and Mathematics, Mindanao State University-Iligan Institute of Technology, Iligan City 9200, Philippines; olga.nuneza@g.msuiit.edu.ph; 2Pharmaceutical Research Institute, Albany College of Pharmacy and Health Sciences, Rensselaer, NY 12144, USA; sudha.thangirala@acphs.edu (T.S.); noureldien.darwish@acphs.edu (N.H.E.D.); shaker.mousa@acphs.edu (S.A.M.); 3Hematology Unit, Clinical Pathology Department, Faculty of Medicine, Mansoura University, Mansoura 35516, Egypt

**Keywords:** functional food, angiogenesis, anticancer, cancer, chlorogenic acid, isoquercetin, *Corchorus olitorius*, MTT assay, CAM assay

## Abstract

*Corchorus olitorius* is a common, leafy vegetable locally known as “Saluyot” in the Philippines. Several studies have reported on its various pharmacological properties, such as antioxidant, anti-inflammatory, analgesic, and anticancer properties. However, little is known about its effects on angiogenesis. This study aimed to evaluate the anticancer properties, such as the antiproliferative, anti-angiogenic, and antitumor activities, of the *C. olitorius aqueous* extract (CO) and its bioactive compounds, chlorogenic acid (CGA) and isoquercetin (IQ), against human melanoma (A-375), gastric cancer (AGS), and pancreatic cancer (SUIT-2), using in vitro and in ovo biological assays. The detection and quantification of CGA and IQ in CO were achieved using LC-MS/MS analysis. The antiproliferative, anti-angiogenic, and antitumor activities of CO, CGA, and IQ against A-375, AGS, and SUIT-2 cancer cell lines were evaluated using MTT and CAM assays. CGA and IQ were confirmed to be present in CO. CO, CGA, and IQ significantly inhibited the proliferation of A-375, AGS, and SUIT-2 cancer cells in a dose-dependent manner after 48 h of treatment. Tumor angiogenesis (hemoglobin levels) of A-375 and AGS tumors was significantly inhibited by CO, CGA, IQ, and a CGA–IQ combination. The growth of implanted A-375 and AGS tumors was significantly reduced by CO, CGA, IQ, and a CGA–IQ combination, as measured in tumor weight. Our investigation provides new evidence to show that CO has promising anticancer effects on various types of human cancer cells. CO and its compounds are potential nutraceutical products that could be used for cancer treatment.

## 1. Introduction

Cancer remains one of the leading causes of death worldwide. One mechanism involved in cancer development is angiogenesis, which is the formation of new blood vessels. Angiogenesis is a normal biological process that plays a pivotal role during embryonic development, wound healing, and the menstrual cycle [1,2,3,4]. However, under pathological conditions, it partakes in an array of diseases in addition to cancer, such as diabetic retinopathy, peripheral vascular disease, endometriosis, tissue regeneration, atherosclerosis, obesity, and rheumatoid arthritis [5]. 

During tumor growth, angiogenesis ensures the supply of sufficient oxygen and fundamental nutrients to cancer cells and tissues [6,7]. Angiogenesis is regulated by various pro-angiogenic factors such as vascular endothelial growth factor (VEGF), platelet-derived growth factor (PDGF), basic fibroblast growth factor (bFGF), transforming growth factor-beta (TGF-β), angiogenin, and interleukin-8 (IL-8). These factors promote endothelial cell stimulation, proliferation, and invasion, resulting in the excessive growth of new capillaries in the tumor [8,9]. VEGF is one of the most important regulators of angiogenesis [10]. It is secreted from cancer and stromal cells in the tumor microenvironment to promote the growth and migration of vascular endothelial cells and the permeability of blood vessels [11].

*Corchorus olitorius* Linn, family Malvaceae, is a common leafy vegetable locally known as “Saluyot” in the Philippines. It is considered by the Philippines Department of Science and Technology [12] as a functional food that provides health benefits beyond an essential nutrient function, such as for reducing the occurrence of lifestyle-related diseases including obesity, atherosclerosis, cardiovascular disease, diabetes, and cancer. Traditionally, *C. olitorius* has been considered a folk remedy for different illnesses such as diabetes, hypertension [13], aches, pains [14], infertility [15], pimples, wounds, boils, insect bites, and swellings. Original research articles have reported the different pharmacological properties of *C. olitorius*, which include antioxidant [16], anti-inflammatory [17], antihyperglycemic, and anticancer properties [18]. 

Polyphenolic compounds are a component of the human diet, and they have various helpful pharmacological properties that have low toxicity [19]. Previous studies have shown that *C. olitorius* leaves contain chlorogenic acid and isoquercetin [20]. Chlorogenic acid is one of the most common polyphenols in the human diet and exhibits many health-promoting benefits [21], such as antioxidant, anti-inflammatory [22], and anticancer benefits [23,24,25,26,27]. Moreover, isoquercetin, also known as isoquercitrin, is a naturally available quercetin glycoside [28,29,30]. In vitro and in vivo studies of glioma, colon, breast, ovarian, and liver cancer showed that isoquercetin exhibits potent, antiproliferative properties [30,31]. However, the mechanisms of action of these compounds as antitumor agents remain unclear.

In this study, bioactive compounds of *C. olitorius* aqueous leaf extract (CO), namely chlorogenic acid (CGA) and isoquercetin (IQ), were qualitatively and quantitatively identified using LC-MS/MS analysis. The effects on cancer cell proliferation were determined using MTT assay of human melanoma (A-375), gastric (AGS), and pancreatic (SUIT-2) cell lines. The anti-angiogenesis and antitumor activities of CO, CGA, and IQ were also explored in the chick chorioallantoic membrane (CAM) model via A-375 and AGS tumor cell implantation.

## 2. Results

### 2.1. Identification and Quantification of Chlorogenic Acid and Isoquercetin

We first identified and quantified the CGA and IQ contents present in 1 mg/mL of CO via chromatographic profiling before conducting the bioassays. Using LC-MS/MS analysis, the presence of CGA (556 ng/mL) and IQ (168 ng/mL) in the plant extract was confirmed and quantified by comparing their chromatograms (Figure 1A) to the chromatograms of CGA and IQ standards (Figure 1B). 

### 2.2. CO, CGA, and IQ Inhibited Human Cancer Cells’ Proliferation

SUIT-2, A-375, and AGS cell lines were selected to assess the cytotoxicity activities of CO, CGA, and IQ while using the MTT assay. As shown in Figure 2, these cell lines treated with various concentrations of CO and CGA had significantly (*p* < 0.01) inhibited cell proliferation after 48 h. By increasing the concentration of CO and CGA, the difference between the cell inhibition percentages of the cell lines were also increased. IQ displayed significantly (*p* < 0.01) strong antiproliferative activity against SUIT-2 cells at the lowest concentration (30 µM) compared with A-375 and AGS cells, which showed significant (*p* < 0.01) effects at 125 µM.

The median inhibitory concentrations (IC_50_) of CO, CGA, and IQ in the cell lines studied are summarized in Table 1. To inhibit 50% of the total cell growth of AGS, A-375, and SUIT-2 cancer cell lines, the concentration of CO must be 2.54, 4.05, and 6.47 mg/mL, respectively. Statistically, CO showed significant antiproliferative activities against AGS (*p* < 0.001) and A-375 (*p* < 0.01) cancer cells. CGA, on the other hand, had IC_50_ values that ranged from 306.64 to 373.53 µM. No significant difference was observed between the cell lines. Lastly, IQ had a low IC_50_ value (144.45 µM) for SUIT-2, implying that this cell line is more sensitive to IQ than the A-375 and AGS cells. The data show that IQ exhibits significant antiproliferative activity on SUIT-2 (*p* < 0.05) when compared to that on the other cell lines.

### 2.3. CO, CGA, and IQ Inhibit Angiogenesis and Tumor Growth in Ovo

To determine the anti-angiogenic and antitumor activities of CO, CGA, IQ, and the combination of CGA and IQ, tumor implantations of A-375 and AGS cells were done in ovo using the CAM assay. The Hb levels and the weights of implanted A-375 or AGS tumors are presented in Figure 3. All treatments significantly (*p* < 0.001) suppressed angiogenesis of A-375 and AGS tumors when compared to the control. However, only AGS was significantly (*p* < 0.001) inhibited by these treatments when compared to the control for tumor growth. Of note, CO significantly (*p* < 0.001) suppressed A-375 and AGS cancer progression, as expressed by their low Hb levels and tumor weights. 

## 3. Discussion

Angiogenesis is a normal biological process that involves the formation of new blood vessels. This process plays an essential role in tumor growth by ensuring sufficient oxygen and nutrient supplies to cancer cells and tissues [6,7]. More often, tumors develop in tissues with dense lymphatic vascular networks and exploit pre-existing lymphatic vessels for invasion and metastasis [32]. Metastasis is considered a primary cause of mortality in cancer patients. Cancer cells break away from the primary tumor during this cellular process and move via blood or lymphatic vessels to proliferate and colonize lymph nodes and distant organs, forming new tumors [33]. Consequently, angiogenesis takes place to support the growing tumor. 

One approach to finding agents to prevent cancer cell proliferation and tumor formation is to study natural agents sourced from nature. In Asia, *C. olitorius* is a common leafy vegetable widely used in preparing dishes [34] and traditional medicine [13,14]. Pharmacological studies have shown that *C. olitorius* exhibits promising antioxidant and anti-inflammatory activities [35] beneficial for treating different cancers [18]. Therefore, in this study, we evaluated the anticancer properties of CO and its bioactive compounds, namely CGA and IQ, against various human cancer cell lines. Our data revealed that CO has significant antiproliferative effects on SUIT-2, A-375, and AGS cells at a concentration as low as 2.54 mg/mL.

Moreover, the results of the in ovo analysis showed that CO strongly inhibited angiogenesis and the growth of A-375 and AGS tumors. CO significantly prevented the formation of new blood vessels of tumors at 10 mg per CAM by decreasing the supply of oxygen and essential nutrients they require. Due to limited oxygen and nutrient supplies, tumor growth of A-375 and AGS were effectively suppressed. Reports have shown that *C. olitorius* can activate procaspases-3 and 9, leading to a cleavage of the downstream substrate, poly (ADP-ribose) polymerase, followed by a downregulation of caspase-activated DNase signaling inhibition [36]. In a genotoxicity study, CO increased DNA damage in the nuclei of myeloma (ARH-77) cells [37]. These anticancer activities can be associated with the ability of *C. olitorius* to induce fragmentation of cancer DNA coupled with nuclear condensation-mediated apoptosis [36].

Consequently, nutraceutical products of *C. olitorius*, including CGA, also showed potent anticancer effects in human cancer. The data gathered in our study showed that 30 µM CGA significantly prevented cancer cell proliferation after 48 h of incubation. Reports on the antiproliferative properties of CGA revealed that it could disrupt the cancer cell cycle and induce apoptosis [38,39,40] via increasing the expression of both Bax and caspase-3 genes and decreasing the level of B-cell lymphoma 2 (Bcl-2) [39]. Additionally, data gathered from CAM models showed that CGA has anti-angiogenic and antitumor properties against A-375 and AGS tumors. CGA at 30 µg/CAM significantly reduced the Hb level of the tumors by inhibiting the process of angiogenesis during eight days of treatment. However, the statistically significant antitumor activity of CGA was only observed in AGS tumors. Recent studies have shown that CGA can inhibit the cell growth of esophageal squamous cell carcinoma (ESCC), lung (A549), liver (AH109A), and kidney (A-498) cancer cell lines both through in vitro and in vivo models [40,41,42,43]. Zhan et al. reported that CGA could downregulate both BMI-1 oncogene and SOX-2 transcription factors in tumor tissues of ectopic xenograft tumors and carcinogen-induced ESCC models [41]. CGA was also found to induce apoptosis by upregulating the expression of both pro-apoptotic Bax and caspase-3 proteins and downregulating the level of the anti-apoptotic Bcl-2 protein in kidney (A-498) and lung (A549) cancer cells [39,40].

IQ, a common flavonoid found in vegetables, is considered a good alternative for preventive therapy, or for supplementing the treatment of various diseases. In this study, data revealed that 30 µM IQ significantly prevented cell proliferation of SUIT-2 after 48 h of incubation. IQ was also assessed to determine its antiangiogenic and antitumor effects on human melanoma and gastric cancer. Our results showed that IQ at 30 µg/CAM efficiently suppressed the angiogenesis and tumor growth of A-375 and AGS cancers implanted in CAMs. In 2014, a study conducted by Hara et al. showed that IQ successfully inhibited the growth of transplanted liver tumors in nude mice [44]. The same study found that IQ suppressed piperonyl butoxide-induced tumor promotion, potentiating PTEN/Akt and disrupting TGF-β/Smad signaling pathways. Several studies have also revealed that IQ can activate caspase −3, −8, and −9, thereby reducing the mitochondrial membrane potential of cancer cells [44,45]. IQ was found to inhibit the expression of ERK and p38 MAPK protein phosphorylation and reduce the expression level of protein kinase C (PKC), leading to the suppression of transplant liver tumors in nude mice [46]. 

Plant extracts are composed of different bioactive compounds. Thus, we combined CGA and IQ with a 1:1 ratio of 10 µg of each compound per CAM to determine their interactions and influences on the anticancer properties of CO against A-375 and AGS tumors. Our data revealed that the combination of CGA and IQ provides synergistic anti-angiogenic and antitumor effects on both tumors. As shown by the low Hb levels of the tumors, the combination of CGA and IQ can significantly suppress the proliferation of new blood vessels, leading to the inhibition of tumor growth. Additionally, it was shown that the combination of CGA and IQ significantly contributed to the anticancer properties of CO. Henceforth, there remains a need to further elucidate the anticancer properties of the remaining compounds present in CO.

## 4. Materials and Methods

### 4.1. Plant Collection and Authentication

*C. olitorius* Linn leaves were collected at Sitio Basak in the Municipality of Tupi, South Cotabato, the Philippines. Plant samples were sent to the Bureau of Plant Industry (BPI) at the Department of Agriculture in Manila City, the Philippines, for proper authentication by a plant taxonomist. Legal documents such as export and local transport permits were secured from the BPI’s satellite office in General Santos City, the Philippines. Experimentations, laboratory analyses, and documentation were conducted at the Pharmaceutical Research Institute of Albany College of Pharmacy and Health Sciences in Rensselaer City, New York, USA.

### 4.2. Plant Preparation

Approximately 5 kg of *C. olitorius* leaves was removed from their stems, washed with distilled water, and pat-dried using paper towels. The fresh leaves were brought to the Chemistry Analytical and Research Laboratory of the Ateneo de Davao University, Davao City, the Philippines for lyophilization using a HyperCOOL 3055 freeze dryer (Daejeon, South Korea) at −55 °C with 1 × 10^−3^ Pa and grinding using an FSJ-1000A swing medicine milling machine (Henan Vic Machinery Co., Ltd., Zhengzhou, China) at 25,000 rpm with a #60 sieve. Pulverized *C. olitorius* leaves were packed into seal-tight plastic bags with packets of silica gels. Bags were stored in a cool and dry place until further use.

### 4.3. Plant Extraction

Fifty grams of *C. olitorius* powder was boiled in distilled water (1 L) for 1 h. After cooling, the plant sample was centrifuged (13,419× *g* for 10 min), and the highly viscous supernatant (gel-like, which suggests the presence of polysaccharides) was collected in a conical flask. The polysaccharides were precipitated by adding an equal volume of ethanol and stirring the sample at around 0 °C using ice. The precipitate was removed by centrifugation (13,419× *g* for 10 min), and the hydroalcoholic supernatant was recovered, filtered, and evaporated using a rotary evaporator. The viscous *C. olitorius* aqueous extract (CO) was stored in an amber vial and kept at −20 °C until further use.

### 4.4. Solid-Phase Extraction

CO (1 mg/mL) was prepared by dissolving the plant extract in distilled water. A solid-phase extraction vacuum manifold with Waters Sep-Pak C18 Vac Cartridge (Waters, Milford, MA, USA) was used to separate dissolved and suspended compounds present in the *C. olitorius* according to their physical and chemical properties. The column was pre-conditioned using 1 mL methanol followed by 1 mL distilled water. The *C. olitorius* was eluted using distilled water–methanol (1:1 ratio) as eluent. The eluant was collected and analyzed with LC–MS/MS to qualify and quantify its CGA and IQ contents.

### 4.5. LC-MS/MS Analysis

An API-4000 mass spectrometer (Sciex, Framingham, MA, USA), equipped with a Shimadzu UPLC system (Kyoto, Japan), was used for LC–MS/MS analyses. A Kinetex 2.6 μm Polar C18 column (50 × 2.1 mm, Phenomenex, Torrance, CA, USA) was used for reversed-phase separation. Mobile phases were: (A) water containing 0.1% formic acid and 5% acetonitrile, and (B) acetonitrile with 0.1% formic acid. The flow rate was 0.4 mL/min and the gradient was linear, from 5% B to 95% B, for 2.5–3 min. The oven temperature was 40 °C and the injection volume was 5 μL. Electro-spray ionization (ESI) was used in negative MRM mode. CGA and IQ were purchased from Sigma-Aldrich (St. Louis, MO, USA). Mass transitions for phytochemicals were: Q1/Q3: 352.9/190.8 (CGA); 463.0/299.9 (IQ). The operative parameters of the mass spectrometer for CGA and IQ were as follows: decluttering potentials (DP): −45 and −85 V; entrance potentials (EP): −10 V; collision energies (CE): −24 and −36 eV; collision cell exit potential (CXP): −17 V; curtain gas (CUR): 50 psi; gas 1 (GS1, nebulizer gas): 50 psi; gas 2 (GS2, heater gas): 30 psi; ion spray voltage (IS): −4500 V; temperature (TEM): 500 °C; collision activate dissociation (CAD) gas: −12 psi; dwell time: 150 ms. Nitrogen was used for the gases. Standard curves were obtained using standard solutions of CGA with concentrations of 200, 100, 50, 20, 10, and 1 ng/mL and IQ with concentrations of 1000, 500, 250, 100, 50, and 5 ng/mL. The LC–MS/MS methods for CGA and IQ were linear (*r* = 0.9979 and 0.9975) within the ranges of standard solutions.

### 4.6. Cell Culture

Human pancreatic (SUIT-2) (MD Anderson Cancer Center, Houston, TX, USA), melanoma (A-375), and gastric (AGS) cancer cells (ATTC, Manassas, VA, USA) were maintained in Roswell Park Memorial Institute 1640 (RPMI-1640) growth medium supplemented with 10% fetal bovine serum, 100 IU/mL penicillin, 100 mg/mL streptomycin, and 2 mM L-glutamine purchased from Sigma-Aldrich. Cultures were maintained in a 37 °C humidified chamber with 5% CO_2_. Media were changed every three days, and the cell lines were passaged at 80% confluence.

### 4.7. 3-(4, 5-Dimethyl thiazolyl-2)-2, 5-Diphenyltetrazolium Bromide (MTT) Assay

Briefly, SUIT-2, A-375, and AGS cancer cells were seeded at a density of 10^4^ cells/well in 96-well plates and incubated for 24 h at 37 °C. After removing the cell culture media, cells were exposed to different concentrations of CO (0 to 400 µg/mL) and its bioactive compounds, CGA and IQ (0 to 500 µM) prepared in the RPMI-1640 media. The cells were then incubated for 48 h, and cytotoxicity was assessed using tetrazolium salt reduction (MTT assay) [47]. Treatments were completely aspirated from each well and replaced with a 110 µL MTT solution (0.5 mg/mL). The plates were covered with tin foil and incubated for another 4 h at 37 °C. The formazan crystals produced by living cells were solubilized by removing 80 µL of MTT solution and by adding 50 µL of dimethyl sulfoxide (DMSO) in each well. The optical density of each well was determined at 570 nm using an ELISA microplate reader. Untreated cells were used as a negative control. The cell viability percentage was calculated by considering the negative control as 100%. All the experiments were performed in triplicate. 

### 4.8. Tumor Growth and Angiogenesis in the Chorioallantoic Membrane Model

One-day-old chick embryos were purchased from Charles River (Norwich, CT, USA) and maintained at 37 °C and 55% relative humidity. After seven days of incubation, the CAM assay was performed, as previously described [48]. In brief, a hypodermic needle was used to make a small hole in the blunt end of the eggshell, with a second hole then made on the broad side of the egg over an avascular portion of the embryonic membrane. Slight suction was applied to the first hole to displace the air sac and drop the CAM away from the shell. Using a Dremel drill (Racine, WI, USA), a window (1 cm^2^) was cut in the shell over the false air sac, allowing access to the CAM. Either A-375 or AGS cells (1 × 10^6^) were mixed in Matrigel and implanted on the CAM. Treatments were applied: CO (10 mg), CGA (30 μg), IQ (30 μg), CGA–IQ combination (10 μg:10 μg), and an integrin αvβ3 antagonist, XT199 (positive control). The antitumor activities of the treatments on tumor angiogenesis and growth were determined 8 days later. A-375 and AGS tumors were extracted by cutting tumors from the CAM and placing them in Eppendorf tubes. Each tumor was weighed using an analytical balance.

Tumor hemoglobin (Hb) content was indexed as a measure of tumor vascularity [32]. In brief, tumor sections were homogenized for 5–10 min in double-distilled water. The samples were centrifugated at 2147× *g* for 10 min, and the supernatant was used for Hb analysis. Drabkin’s reagent was added to the supernatant, and Hb absorbance was measured at 540 nm with an ELISA microplate reader. Hb concentration was expressed as mg/mL based on comparison with a standard curve. Data represent the mean ± SEM of tumor weight (in grams) per treatment group and tumor Hb (in mg/dL), where *n* = 5 per group.

## 5. Conclusions

This study is the first to establish that CO and its bioactive compounds CGA and IQ can inhibit the proliferation of various human cancer cells in vitro and suppress the growth of A-375 and AGS tumor xenografts in ovo in a CAM model by inducing apoptosis and anti-angiogenetic properties. Taken together, the anticancer activities of CO and its bioactive compounds can be associated with their capacity to regulate various proteins and signaling pathways in order to induce DNA damage in cancer cells. Therefore, CO, CGA, and IQ may have the potential to be developed clinically as novel anticancer agents.

## Figures and Tables

**Figure 1 molecules-26-06033-f001:**
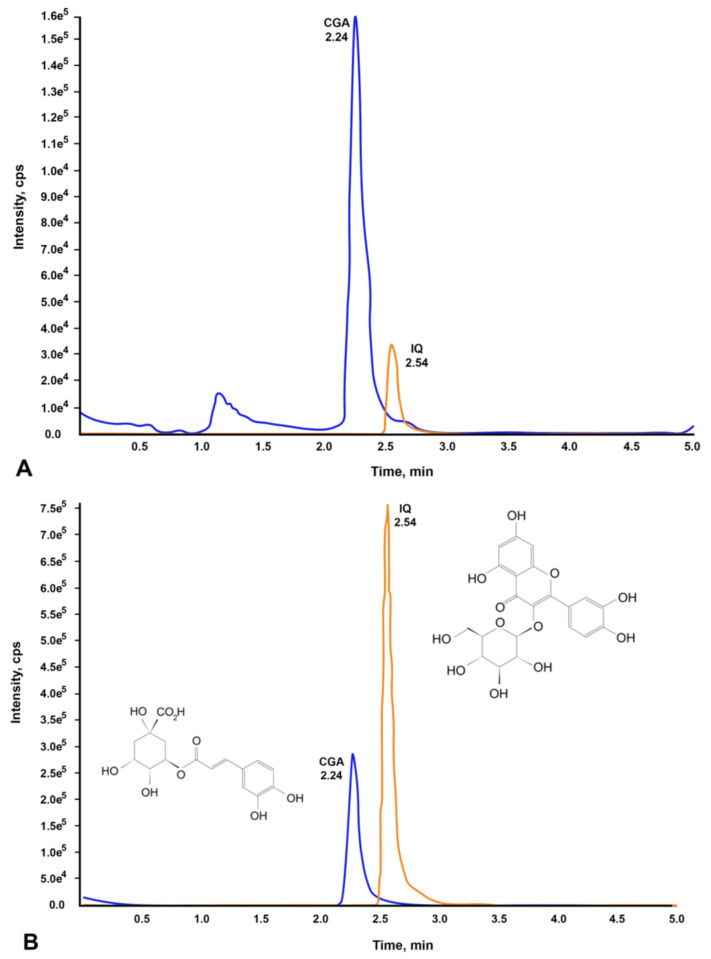
Chromatograms of CGA and IQ found in *C. olitorius* aqueous extract. (**A**) CGA and IQ present in CO, and (**B**) standard compounds (CGA and IQ).

**Figure 2 molecules-26-06033-f002:**
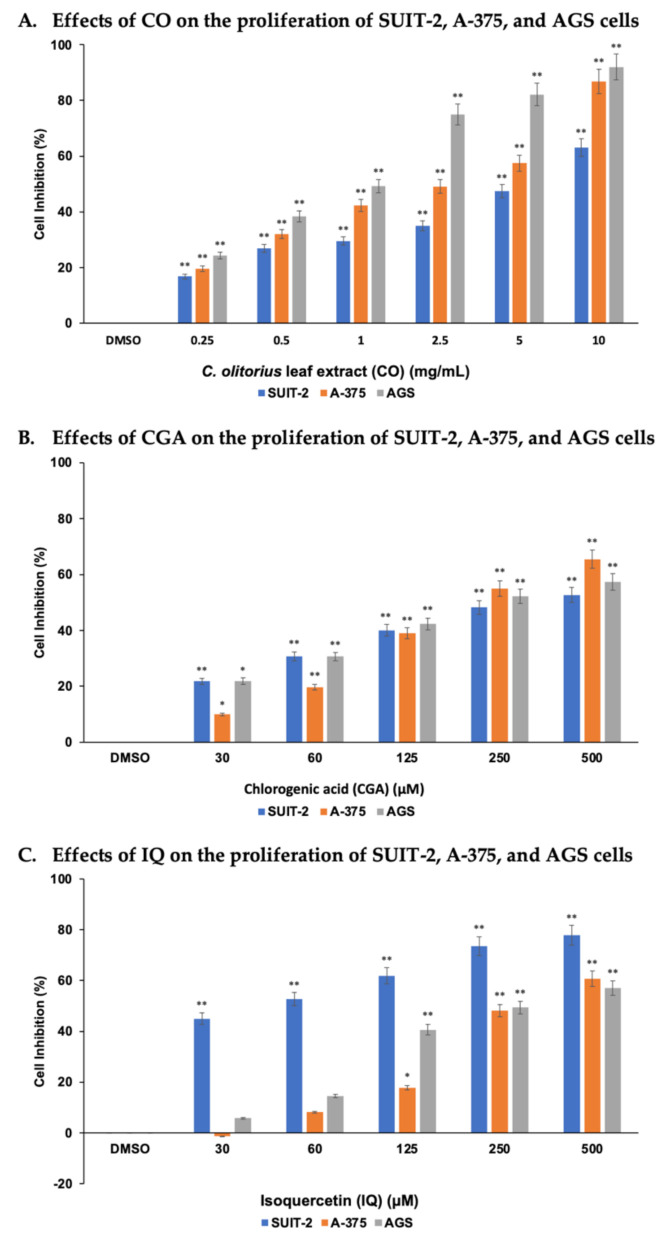
CO, CGA, and IQ induced antiproliferative activities in human cancer cells in the MTT assay (**A**–**C**) after 48 h. Data are represented as mean ± SEM (*n* = 3). * *p* < 0.05 or ** *p* < 0.01 as compared to 0.5% DMSO treatment (control group).

**Figure 3 molecules-26-06033-f003:**
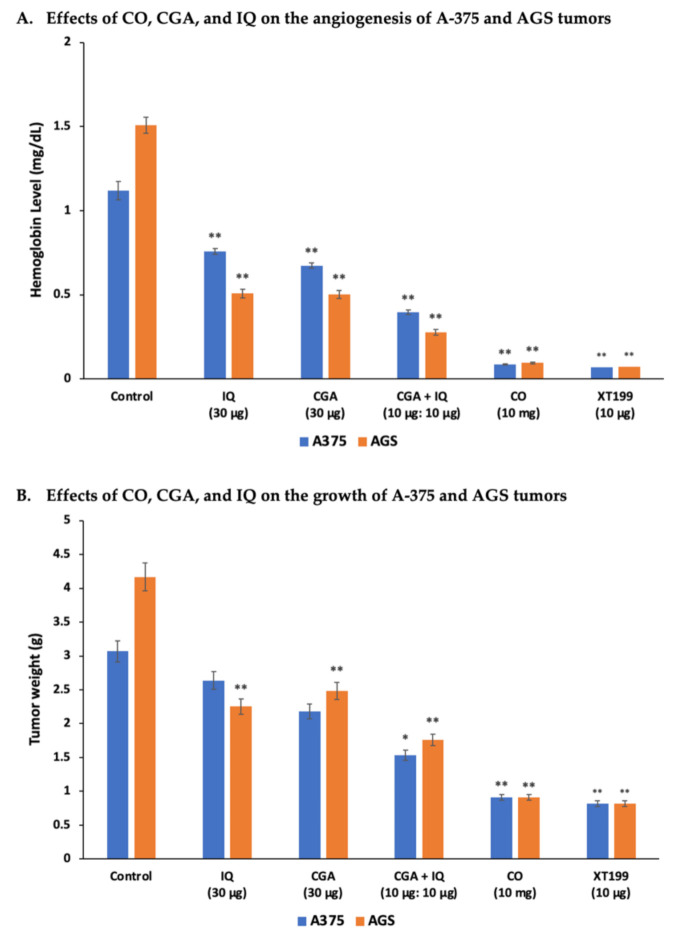
CO, CGA, IQ, and CGA + IQ inhibited angiogenesis and tumor growth of A-375 and AGS cancer cells. (**A**) Hemoglobin level measured from the CAM-implanted tumors after eight days. (**B**) Weight of the A-375 and AGS CAM tumors eight days after implantation and treatment. Data are represented as mean ± SEM (*n* = 3). * *p* < 0.01, ** *p* < 0.001 versus the control (untreated) group.

**Table 1 molecules-26-06033-t001:** Median inhibitory concentration (IC_50_) values of *Corchorus olitorius* aqueous extract, chlorogenic acid, and isoquercetin for SUIT-2, A-375, and AGS cancer cell lines.

Treatment	IC_50_ Values		
	SUIT-2	A-375	AGS
CO (mg/mL)	6.47 ± 0.11 ^##^	4.05 ± 0.09 **	2.54 ± 0.09 ***
CGA (µM)	373.53 ± 11.29 ^ns^	306.64 ± 20.83 ^ns^	330.74 ± 13.19 ^ns^
IQ (µM)	144.45 ± 12.34 ^#^	369.29 ± 23.78 *	355.97 ± 19.93

The significance of the mean differences was assessed using one-way ANOVA. * *p* < 0.05, ** *p* < 0.01, and *** *p* < 0.001 compared with SUIT-2 cells. ^#^
*p* < 0.05 and ^##^
*p* < 0.01 compared with A-375. ^ns^ No significant difference between groups.

## Data Availability

Not applicable.

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
