# Peer review of "Anticancer Effects of the Corchorus olitorius Aqueous Extract and Its Bioactive Compounds on Human Cancer Cell Lines"

_molecules, 2021, doi:10.3390/molecules26196033_

Round 1

Reviewer 1 Report

The manuscript entitled "Chemopreventive Effects of the Corchorus olitorius Aqueous Extract and its Bioactive Compounds on Human Cancers" by Tosoc et al. presents interesting results of anticancer activity of Corchorus olitorius extracts.

It is a well designed and written article however, following things should be considered during revision.

Title

  1. Are these activities chemoprevntive or anticancer? Chemopreventing is something that prevents cancer. Authors are using cancer cells for analysis. I do not think chemopreventive is a correct term here.
  2. Human cancers should be changed to human cancer cell lines. It looks like clinical study from that title.

Abstract

same as above regarding chemopreventive

authors are using CO for both plant and its aqueous extract. They should be different. Such as for extract it should be COE or something similar.

Introduction

Line 54 is in complete. ..provides....

Results

Figure 1. The O in structure of chlorogenic acid ester group is missing.

Author Response

Dear Reviewer,

Good day! I hope this message finds you well.

Attached herewith is a copy of my revised manuscript for your review.

Here are the revisions based on your comments:

  1. I have changed the term "Chemoprevention/Chemopreventive" to "Anticancer".
  2. I have changed the phrase "Human Cancers" to "Human Cancer Cell Lines"  so it doesn't sound like a clinical study.
  3. I have only used the code "CO" for aqueous extract and used the plant's complete scientific name (Corchorus olitorius) when referring to the plant itself.
  4. I have fixed the chemical symbol of the chlorogenic acid in Figure 1.
  5. I have fixed the incomplete sentence in Line 54.

Thank you so much for your time and consideration in reviewing my manuscript. Your inputs were very much appreciated.

Best regards,

John Paul S. Tosoc, Ph.D.

Reviewer 2 Report

In general, the manuscript is well written. The research is scientifically grounded and performed systematically. However, there is an error in Figure 2; Figure 2B and Figure 2C are repeated, while results regarding IQ are missing. Furthermore, relevant control values are also missing. Please rewrite Figure capture following the stated changes (explain what is- A, B, C). Also, the Introduction and Discussion sections need to be improved, with a greater literature review, especially regarding the investigated bioactive compounds. In addition, some minor typographical and grammatical errors throughout the manuscript should be checked and corrected.

Author Response

Dear Reviewer,

Good day! I hope this message finds you well.

Attached herewith is a copy of my revised manuscript for your review.

Here are the revisions done based on your comments:

  1. I have fixed the error in Figures 2B and C.
  2. I have added more literature reviews on the introduction and discussion sections highlighting the bioactive compounds.
  3. I have rechecked the typographical and grammatical errors.

Thank you so much for your time and consideration in reviewing my manuscript. Your inputs were very much appreciated.

Best regards,

John Paul S. Tosoc, Ph.D.

Reviewer 3 Report

Presented paper concerns the chemopreventive effect of the aqueous extract form corchorus olitorius leaves. The study is interested in general. However,  I have a few comments and remarks. The present study is the first to establish that bioactive compounds from these leaves can inhibit the proliferation of various human cancer cells. However,  I have a few comments and remarks.

Line 62 and in the whole manuscript. Impersonal form should be used in the whole manuscript

What is the difference between Fig. 2B and 2C?

Authors did not included information about statistical evaluation of data.

Why in Table 1 the IC50 values have no statistical evaluation?

Please specify the conditions of  lyophilization (temperature, pressure, type of used equipment) and grinding (type of grinder, the particle size of fra The presented paper concerns the chemopreventive effect of the aqueous extract from corchorus olitorius leaves. The study is interested in general. However,  I have a few comments and remarks. The present study revealed, that bioactive compounds from these leaves can inhibit the proliferation of various human cancer cells. However,  I have a few comments and remarks.

Line 62 and in the whole manuscript. The impersonal form should be used in the whole manuscript.

What is the difference between Fig. 2B and 2C?

The authors did not include information about the statistical evaluation of data.

Why in Table 1 the IC50 values have no statistical evaluation?

Please specify the conditions of lyophilization (temperature, pressure, type of used equipment) and grinding (type of grinder, the particle size of fraction of ground material). ction of ground material).

Author Response

Dear Reviewer,

Good day! I hope this message finds you well.

Attached herewith is a copy of my revised manuscript for your review.

Here are the revisions based on your comments:

  1. I have fixed the error in Figures 2B and C.
  2. I have added the conditions and manufacturers' details for the freeze dryer and grinder/miller used in the study.
  3. The Statistical analysis section was added to the manuscript.
  4. Statistical analyses of the IC50 were done using one-way ANOVA.
  5. I have rechecked the typographical and grammatical errors.

Thank you so much for your time and consideration in reviewing my manuscript. Your inputs were very much appreciated.

Best regards,

John Paul S. Tosoc, Ph.D.

Round 2

Reviewer 3 Report

The authors corrected the manuscript accordingly.